# CONTINUAL BACKPROP: STOCHASTIC GRADIENT DESCENT WITH PERSISTENT RANDOMNESS

## ABSTRACT

The Backprop algorithm for learning in neural networks utilizes two mechanisms: first, stochastic gradient descent and second, initialization with small random weights, where the latter is essential to the effectiveness of the former. We show that in continual learning setups, Backprop performs well initially, but over time its performance degrades. Stochastic gradient descent alone is insufficient to learn continually; the initial randomness enables only initial learning but not continual learning. To the best of our knowledge, ours is the first result showing this degradation in Backprop's ability to learn. To address this issue, we propose an algorithm that continually injects random features alongside gradient descent using a new generate-and-test process. We call this the *Continual Backprop* algorithm. We show that, unlike Backprop, Continual Backprop is able to continually adapt in both supervised and reinforcement learning (RL) problems. We expect that as continual learning becomes more common in future applications, a method like Continual Backprop will be essential where the advantages of random initialization are present throughout learning.

## 1 INTRODUCTION

In the last decade, deep learning methods have been successful and become the state-of-the-art in many machine learning problems and applications, including supervised classification, reinforcement learning (Silver et al., 2016), computer vision (Krizhevsky et al., 2012), and natural language processing (Brown et al., 2020). These methods learn the weights of an artificial neural network using Backprop, which is primarily applied to stationary problems. However, a primary challenge to leveraging the strengths of deep learning beyond current applications is that Backprop does not work well in non-stationary problems (McCloskey and Cohen, 1989; French, 1997; Sahoo et al., 2018), for example, a problem that consists a sequence of stationary problems.

Leveraging the strengths of deep learning methods in non-stationary problems is important as many real-world applications of machine learning like robotics involve non-stationrities. Non-stationarities can arise due to changes in the environment (Thurn, 1998), high complexity (Sutton et al., 2007), partial observability (Khetarpal et al., 2020), or other actors (Foerster et al., 2018). Many works have tried to make deep learning work in non-stationary problems. Some works have proposed methods that can remember previously learned information (Kirkpatrick et al., 2017; Aljundi et al., 2018; Riemer et al., 2019). While, others have proposed methods that can adapt fast to non-stationarities (Rusu et al., 2016; Al-Shedivat et al., 2018; Finn et al., 2019).

A limitation of prior works on non-stationary problems is that they did not study cases where there are many non-stationarities. Number of non-stationarities refers to the number of times the data distribution changes. Most works only look at problems with less than ten non-stationarities (Rusu et al., 2016; Kirkpatrick et al., 2017; Al-Shedivat et al., 2018). Finn et al. (2019) studied problems with up to a hundred non-stationarities. Dealing with many non-stationarities is important for systems that continually interact with the real world as non-stationarities frequently occur in the world.

In this work, we study problems with a large number (thousands) of non-stationarities. We start with a special class of problems that we call *semi-stationary* problems. These are online supervised learning problems where the input distribution is non-stationary while the target function is stationary. The target function is the function being approximated, for example, the true regression function. These problems are a natural way to study problems with many non-stationarities; slowly

moving through a large input space can cause thousands of non-stationarities. These problems are also important for systems in the real world as inputs from the real world often depend on previous inputs, making the input distribution non-stationary. Finally, we study non-stationary RL problems. These are full non-stationarity problems as the input distribution changes when the agent's behaviour changes and the target function, optimal policy, changes as the dynamics of the environment change.

Semi-stationary problems reveal a new difficulty with Backprop(BP) in non-stationary problems; they help us clearly understand one way in which Backprop fails. We show that in non-stationary problems, Backprop performs well initially, but surprisingly, its performance degrades substantially over time as Backprop loses its ability to adapt. Backprop relies on proper random initialization for its effectiveness (Glorot et al., 2010; Sutskerver et al., 2013; He et al., 2015). However, randomization in Backprop only happens in the beginning. We hypothesize that Backprop's ability to adapt degrades because the benefits of initial random distribution are not present at all times.

To extend the benefits of initial random distribution throughout learning, we propose the Continual Backprop (CBP) algorithm. CBP uses a generate-and-test method to continually inject random features alongside SGD. We show that unlike BP, CBP can continually adapt in non-stationary problems. Our generate-and-test algorithm consists of two parts: the generator, which proposes new features,and the second is the tester, which finds and replaces low utility features with the features proposed by the generator. Our generate-and-test algorithm is built on the one proposed by Mahmood and Sutton, (2013) and our algorithm is compatible with modern deep learning. We overcome three significant limitations of their work. First, our algorithm is applicable to feed-forward networks with arbitrary shape and size while theirs was only applicable to networks with a single hidden layer and one output. Second, our algorithms works with modern activations and optimizers like Adam while theirs was limited to LTU activations, binary weights, and SGD. And third, we combine generate-and-test with SGD and show that the resulting algorithm, CBP, is significantly better than BP in complex non-stationary problems while their work was an isolated study of generate-and-test.

The first contribution of our work is to show that in non-stationary problems with many non-stationarities BP loses its ability to adapt over time. In other words, we contribute to the understanding of why BP and its variants fails in non-stationary problems. Our second contribution is that we propose the CBP algorithm that extends the benefits of the initialization in BP to all times.

## 2 NON-STATIONARY PROBLEMS

We study Backprop, Continual Backprop, and other learning algorithms in semi-stationary and Reinforcement Learning (RL) problems. First, we consider a novel idealized semi-stationary problem. The strength of this problem is that in this problem we can study continual learning algorithms extensively and yet in a computationally inexpensive way and without the confounders that arise in more complex problems. Then we study the permuted MNIST problem, an online image classification problem, and two non-stationary RL problems. We demonstrate on these problems that the findings from the idealized problem scale to large neural networks in more realistic settings.

**Performances measure in semi-stationary problems**

In supervised learning, the task is to learn a function using examples of input-output pairs. This function is called the target function. In online supervised learning (Orabona, 2019), there is a stream of samples $(x_t, y_t)$, and the predictions have to be made sequentially. The performance measure is the loss on the next sample. Thus, learning and evaluation happen simultaneously. This is fundamentally different from offline supervised learning, where there are two separate phases, one for learning and another for evaluation. Another common measure in non-stationary problems is the performance on previously seen data. However, measuring performance on previously seen data is only meaningful when studying the catastrophic forgetting aspect of BP. As we do not study the forgetting problem, we do not measure the performance on old data.

### 2.1 BIT-FLIPPING PROBLEM

Our first problem is the *Bit-Flipping problem*. It differs slightly from most supervised learning in two ways. First, it is conventionally assumed that samples are independently and identically distributed, whereas we focus on the case where the sample at the current time-step depends on the previous

sample. Second, it is often assumed that the learner has sufficient capacity to closely approximate the target function, whereas we assume that the target function is more complex than the learner. The best approximation continually changes in this problem as the input distribution is non-stationary, and the target function has high complexity. Therefore, there is a need for continual adaptation.

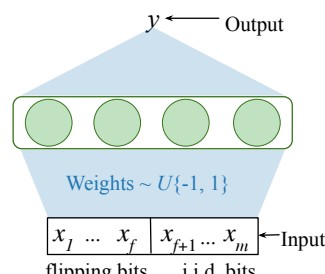

The target function in the Bit-Flipping problem is represented by a multi-layered *target network*. The target network has two layers of weights. We limit the learning networks to networks with the same depth. This allows us to control the relative complexity of the target function and learner. If the target network has a lot more hidden units than the learner, then the target function is more complex than the learner. We set the target network to be wider.

The input at time step $t$, $\mathbf{x}_t$, is a binary vector of size $m$. Here, $\mathbf{x}_t \in \{0,1\}^m$ where each element, $x_{i,t} \in \{0,1\}$ for $i$ in $1,...,m$. After every $T$ time-steps, one of the first $f$ bits is randomly selected and its value is flipped; the values of these $f$ bits do not change at other times. We refer to the first $f$ bits as *flipping bits*. Each of the next $m-f$ bits is randomly sampled from $U\{0,1\}$ at every time-step. The value of $T$ allows us to control the correlation among the consecutive values of the input. Note that when a flipping bit flips, the input distribution changes. We use $m=20$, $f=15$, and $T=1e4$.

In the target network, all weights are randomly sampled from $U\{-1,1\}$. The activation function is a Linear Threshold Unit (LTU), McCulloch (1943). The output of an LTU, with input $\mathbf{x}_t$ is 1 if $\sum_{i=0}^{m+1} v_i x_{i,t} > \theta_i$ else 0. Here, $\mathbf{v}$ is the input weight vector. We set $\theta_i = (m+1) * \beta - S_i$, where $S_i$ is the number of input weights with the value of $-1$ and $\beta \in [0,1]$. This form of LTU is taken from Sutton and Whitehead, (1994). We use $\beta = 0.7$. The output of the network at time-step $t$ is a scalar $y_t \in \mathbb{R}$. Figure 1 shows the input and the target network. The Bit-Flipping problem is a regression problem; we use the squared error to measure performance.

Figure 1: The input and target function generating the output in the Bit-Flipping problem. The input has $m+1$ bits. One of the flipping bits is chosen after every $T$ time-steps, and its value is flipped. The next $m-f$ bits are i.i.d. at every time-step. The target function is represented by a neural network with a single hidden layer of LTUs.

## 2.2 PERMUTED MNIST

We use an online variant of the Permuted MNIST problem (Zenke et al., 2017). They used this problem with just 10 non-stationarities. This is an image classification problem with 10 classes. The images in permuted MNIST are generated from the images in the MNIST dataset by randomly permuting the pixels in the images. We present the images sequentially and measure online classification accuracy. The MNIST dataset has 60,000 images. We present these 60k images in random order, and after all the images have been presented, we use a single permutation to change all the images. This cycle of presenting all the 60,000 images and then changing the permutation of pixels can be continued indefinitely, which allows us to create a long-term continual learning problem.

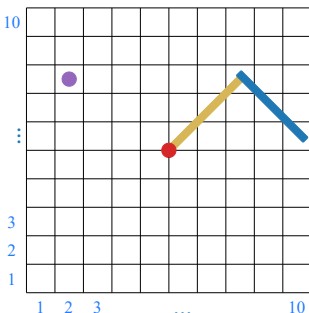

Figure 2: Target selection in Semi-stationary Reacher is a two-step process. First, one of the 100 sub-squares is selected for 30 episodes. Second, for each episode, a new target point is randomly selected inside the current sub-square. Here, the target is selected from sub-square (2,8).

## 2.3 NON-STATIONARY RL PROBLEMS

We use two non-stationary RL problems and both are continual variants of a corresponding Pybullet (Coumans and Bai, 2016) problem. Non-stationary variants are needed as all the problems in Pybullet are stationary. In our problems, either the environment or the distribution of input changes after a pre-specified time, making it necessary for the learner to adapt.

**Slippery Ant:** In this problem, we change the friction between the agent and the ground of the standard Pybullet Ant problem. In the standard problem, the value of the friction is $1.5$. We change the friction after every 10M time-steps by log-uniformly sampling it from $[1e-4, 1e4]$.

**Semi-stationary Reacher:** This problem is a modification of PyBullet's Reacher problem. In Reacher, the goal is to get the tip of an arm to a target point in a square. At the beginning of each episode, a new target point is uniformly randomly chosen inside the square. Each episode lasts for 150 time-steps. In our problem, we change the distribution of input after every 30 episodes (4500 time-steps). We do so by dividing the square into 100 sub-squares, as shown in Figure 2. At the beginning of each episode, the target point is randomly sampled from within a sub-square. This sub-square is changed after every 4500 time-steps when a new sub-square is randomly selected.

## 3 BACKPROP LOSES THE ABILITY TO ADAPT UNDER EXTENDED TRACKING

In the Bit-Flipping problem, the *learning networks* is the network that is used to predict the output. This network has a single hidden layer with 5 hidden units, while the target network has one hidden layer but 100 hidden units. So, the target network is more complex than the learning network. Because the input distribution is changing over time and the target function is more complex than the learner, the best approximation continually changes and thus, there is a need to track the best approximation.

We first used Backprop to track the best approximation. Tracking with Backprop as the learning process in an online supervised learning problem consists of randomly initializing the weights and updating them after every example using the gradient with respect to the loss on the current example. That is, we used Backprop in an incremental manner without any mini-batches.

We studied learning networks with six different non-linear activations: tanh, sigmoid, ReLU (Nair and Hinton, 2010), Leaky-ReLU (Mass et al., 2013) with a negative slope of 0.01, ELU (Clevert et al., 2015), and Swish (Ramachandran et al., 2017). We studied all these networks when they learned using SGD or its variant Adam (Kingma and Ba, 2015).

We used uniform Kaiming distribution (He et al., 2015) to initialize the learning network's weights. The distribution is $U(-b, b)$ with bound, $b = gain * \sqrt{\frac{3}{num\_inputs}}$, where $gain$ is chosen such that the magnitude of inputs does not change across layers. For tanh, Sigmoid, ReLU, and Leaky-ReLU, $gain$ is 5/3, 1, $\sqrt{2}$, and $\sqrt{2/(1+\alpha^2)}$ respectively. For ELU and Swish, we used $gain = \sqrt{2}$, as was done in the original works by Clevert et al. (2016) and Ramachandran et al. (2017).

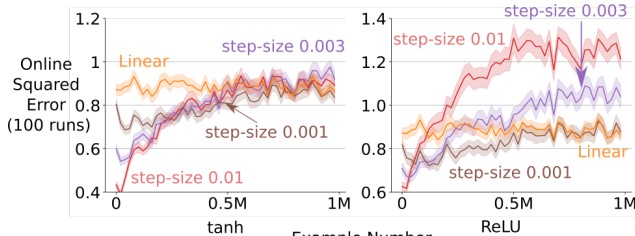

Figure 3: The learning curve on the Bit-Flipping problem using Backprop. Surprisingly, after performing well initially, the error goes up for all step-sizes and activation functions. Backprop's ability to track becomes worse under extended tracking on the Bit flipping problem. For Relu, its performance gets even worse than the linear learner.

We used a linear tracker as a baseline in the Bit-Flipping problem. The linear tracker predicts that output as a linear combination of the inputs. The weights for the linear combination are learned with SGD. We chose the step-size parameter that had least total error.

We ran the experiment on the Bit-Flipping problem for 1M examples. For each activation and value of step-size, we performed 100 independent runs. For different activations and values of the step-size parameter, the same 100 sequences of data (input-output pairs) were used.

Figure 3 shows the squared error of different learning networks with different step-sizes during the first 1M examples. We bin the online error into bins of size 20,000. The shaded region in the figure shows the standard error of the binned error. To display the degradation for step-size of 0.001 more clearly, we ran the experiment for 5M examples, and the results are presented in Appendix A. In Appendix A, we also show similar results when we used Adam instead of SGD.

In the permuted MNIST problem, we used a feed-forward network with three hidden layers with 2000 hidden units and ReLU activations and SGD for updates.

Figures 3 and 10 show that in the Bit-Flipping problem, for all reasonable step-sizes, the error with SGD either increases significantly. Figures 11 and 12 in the appendix show similar results with

Adam. Figure 4 shows that in the permuted MNIST problem, the performance for all steps-sizes gets worse over time. The degradation in performance is slower for smaller step-sizes. From these results, we conclude that Backprop and its variants are not suitable for extended tracking as they continue to lose their ability to adapt over time.

The weights of the learning network are different at the beginning and after a few gradient-based updates. In the beginning, they were small random weights; however, after learning for some time, the weights became optimized to reduce the error for the most recent input distribution. Thus, the starting weights to get to the next solution have changed. As this difference in the weights is the only difference in the learning algorithm across time, the initial weight distribution must have some special properties that allow the learning network to adapt fast. The initial random distribution might have many properties that allow fast adaptation, like diversity of features, non-saturated features, small weight magnitude etc.

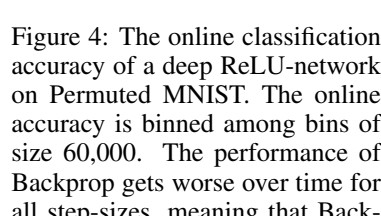

Figure 4: The online classification accuracy of a deep ReLU-network on Permuted MNIST. The online accuracy is binned among bins of size 60,000. The performance of Backprop gets worse over time for all step-sizes, meaning that Backprop loses its ability to adapt under extended tracking.

## 4 CONTINUAL BACKPROP

The benefits of initializing with small random numbers in Backprop are only present initially, as initialization with small random weights only happens at the start. This special initialization makes Backprop temporally asymmetric as it does a special computation at the beginning which is not continued later. But continually learning algorithms should be temporally symmetric as they should do similar computations at all times.

One way to extend the benefits of initialization throughout learning is by continually injecting random features. A feature refers to a hidden unit in a network. However, we are interested in continually learning methods that have similar memory requirements at all times because methods whose memory requirements grow with the amount of data are not scalable. So we also need to remove some features without affecting the already learned function too much. The features to be replaced can be chosen either randomly or using some measure of utility.

Continually replacing low utility features with new random features is a natural way to extend the benefits of initialization to all times while using constant memory. We use a generate-and-test process to continually replace low utility features with new random features. This process has two parts. First is a generator that provides new random features from the initial distribution. The second is a tester that finds low utility features and replaces them with features proposed by the generator. Our generate-and-test process is applicable to arbitrary feed-forward networks, modern activations, and optimizers. While the prior algorithm (Mahmood & Sutton, 2013) was limited to learning networks with a single hidden layer, only one output, LTU activation and could only use SGD to learn.

The generator creates new features by randomly sampling from the distribution that was used to initialize the weights in the layer. When a new feature is added, its outgoing weights are initialized to zero. Initializing as zero ensures that the newly added feature does not affect the already learned function. However, initializing the outgoing weight to zero makes it vulnerable to immediate replacement. The new features are protected for *maturity-threshold*, $m$, number of updates.

The tester finds low utility features and replaces them. At every time-step, *replacement-rate*, $\rho$, fraction of features are replaced in every layer. Our utility measure has two parts: first measures the contribution of the feature to the next features, and second measures features' ability to adapt.

The *contribution-utility* is defined for each connection or weight and for each feature. The basic intuition behind the contribution part is that magnitude of the product of feature activation, and outgoing weight gives information about how useful this connection is to the next feature. If the features contribution to the next feature is small, it can be overwhelmed by contributions from other features. Then the current feature is not useful for the consumer. The same measure of connection utility was proposed by Hu et al. (2016) for the network pruning problem. We define the contribution-utility of a feature as the sum of the utilities of all its outgoing connections. The contribution-utility is measured as a running average with a decay rate, $\eta$. In a feed-forward neural network, the contribution-utility,

$c_{l,i,t}$, of the $i$th features in layer $l$ at time $t$ is updated as

$$c_{l,i,t} = (1 - \eta) * |h_{l,i,t}| * \sum_{k=1}^{n_{l+1}} |w_{l,i,k,t}| + \eta * c_{l,i,t-1}, \qquad (1)$$

where $h_{l,i,t}$ is the output of the $i$th feature in layer $l$ at time $t$, $w_{l,i,k,t}$ is the weight connecting the $i$th unit in layer $l$ to the $k$th unit in layer $l + 1$ at time $t$, $n_{l+1}$ is the number of units is layer $l + 1$.

In the network pruning problem, the connections are removed after learning is done. However, generate-and-test is used during learning, so we need to consider the effect of the learning process on the contribution. Specifically, we need to consider the part of the contribution that will be transferred to the bias when a feature is removed. When a feature is removed, SGD will transfer the average part of the contribution to the bias unit over time. We define, the mean-corrected contribution utility, $z_{l,i,t}$, is the magnitude of the product of connecting weight and input minus its average value,

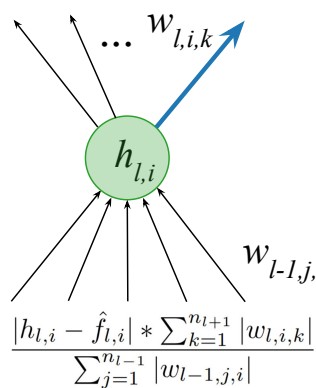

$$f_{l,i,t} = (1 - \eta) * h_{l,i,t} + \eta * f_{l,i,t-1}, \qquad (2)$$

$$\hat{f}_{l,i,t} = \frac{f_{l,i,t-1}}{1 - \eta^{a_{l,i,t}}}, \qquad (3)$$

$$z_{l,i,t} = (1 - \eta) * |h_{l,i,t} - \hat{f}_{l,i,t}| * \sum_{k=1}^{n_{l+1}} |w_{l,i,k,t}| + \eta * z_{l,i,t-1}, \qquad (4)$$

$$\frac{|h_{l,i} - \hat{f}_{l,i}| * \sum_{k=1}^{n_{l+1}} |w_{l,i,k}|}{\sum_{j=1}^{n_{l-1}} |w_{l-1,j,i}|}$$

where $h_{l,i,t}$ is features' output, $w_{l,i,k,t}$ is the weight connecting the feature to the $k$th unit in layer $l + 1$, $n_{l+1}$ is the number of units is layer $l + 1$, $a_{l,i,t}$ is the age of the feature at time $t$. Here, $f_{l,i,t}$ is a running average of $h_{l,i,t}$ and $\hat{f}_{l,i,t}$ is the bias-corrected estimate.

We define the *adaptation-utility* as the inverse of the average magnitude of the features' input weights. The adaptation-utility captures how fast a feature can adapt. The inverse of the weight magnitude is a reasonable measure for the speed of adaptation for Adam-type optimizers. In Adam, the change in a weight in a single update is either upper bounded by the step-size parameter or a small multiple of the step-size parameter (Kingma and Ba, 2015). So, during each update, smaller weights can have a larger relative change in the function they are representing.

Figure 5: A feature/hidden-unit in a network. The utility of a feature at time $t$ is the product of its contribution utility and its adaptation utility. Adaptation utility is the inverse of the sum of the magnitude of the incoming weights. And, contribution utility is the product of the magnitude of the outgoing weights and feature activation ($h_{l,i}$) minus its average ($\hat{f}_{l,i}$). $\hat{f}_{l,i}$ is a running average of $h_{l,i}$.

We define the overall utility of a feature is the product of its contribution-utility and adaptation-utility. The overall utility, $\hat{u}_{l,i,t}$, becomes

$$u_{l,i,t} = (1 - \eta) * \frac{|h_{l,i,t} - \hat{f}_{l,i,t}| * \sum_{k=1}^{n_{l+1}} |w_{l,i,k,t}|}{\sum_{j=1}^{n_{l-1}} |w_{l-1,j,i,t}|} + \eta * u_{l,i,t-1}, \qquad (5)$$

$$\hat{u}_{l,i,t} = \frac{u_{l,i,t-1}}{1 - \eta^{a_{l,i,t}}}. \qquad (6)$$

Figure 5 describes the utility for a feature. Our utility measure is more general than the one proposed by Mahmood and Sutton (2013), as ours applies to networks where features have multiple outgoing connections. In contrast, theirs was limited to the case when features with one outgoing connection.

The final algorithm combines Backprop with our generate-and-test algorithm to continually inject random features from the initial distribution. We refer to this algorithm as *Continual Backprop* (CBP). CBP performs a gradient-descent and a generate-and-test step at every time step. Algorithm 1 specifies the CBP algorithm for a feed-forward neural network. We describe CBP with Adam in Appendix B which contains adam specific details. The name "Continual" Backprop comes from an algorithmic perspective. The Backprop algorithm, as proposed by Rumelhart et al. (1987), had two parts, initialization and gradient descent. However, initialization only happens initially, so Backprop is not a continual algorithm as it does not do similar computation at all times. On the other hand, CBP is continual as it performs similar computation at all times.

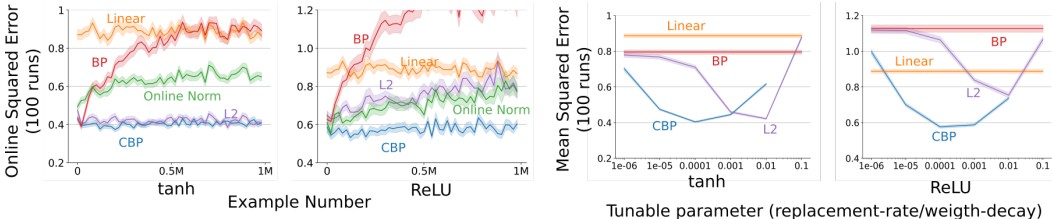

Figure 6: The learning curves and parameter sensitivity plots of Backprop(BP), Backprop with L2, Backprop with Online Normalization, and Continual Backprop (CBP) on the Bit-Flipping problem. Only CBP has a non-increasing error rate in all cases. Continually injecting randomness alongside gradient descent, CBP, is better for continual adaptation than just gradient descent, BP.

---

**Algorithm 1:** Continual Backprop (CBP) for a feed forward neural network with $L$ hidden layers

**Set:** step-size $\alpha$, replacement rate $\rho$, decay rate $\eta$, and maturity threshold $m$ (e.g. $10^{-4}$, $10^{-4}$, 0.99, and 100)
**Initialize:** Initialize the weights $\mathbf{w}_0, ..., \mathbf{w}_L$. Let, $\mathbf{w}_l$ be sampled from a distribution $d_l$
**Initialize:** Utilities $\mathbf{u}_1, ..., \mathbf{u}_L$, average feature activation $\mathbf{f}_1, ..., \mathbf{f}_l$, and ages $\mathbf{a}_1, ..., \mathbf{a}_L$ to 0
**for** *each input $x_t$* **do**
  **Forward pass:** pass input through the network, get the prediction, $\hat{y}_t$
  **Evaluate:** Receive loss $l(x_t, \hat{y}_t)$
  **Backward pass:** update the weights using stochastic gradient descent
  **for** *layer $l$ in $1 : L$* **do**
    **Update age:** $\mathbf{a}_l \mathrel{+}= 1$
    **Update feature utility:** Using Equation 5
    **Find eligible features:** Features with age more than $m$
    **Features to replace:** $n_l * \rho$ of eligible features with smallest utility, let their indices be $\mathbf{r}$
    **Initialize input weights:** Reset the input weights $\mathbf{w}_{l-1}[\mathbf{r}]$ using samples from $d_l$
    **Initialize output weights:** Set $\mathbf{w}_l[\mathbf{r}]$ to zero
    **Initialize utility, feature activation, and age:** Set $\mathbf{u}_{l,\mathbf{r},t}$, $\mathbf{f}_{l,\mathbf{r},t}$, and $\mathbf{a}_{l,\mathbf{r},t}$ to 0

---

### Continual Backprop in semi-stationary problems

The most common ways to keep the weight distribution close to the initial distribution are L2 regularization and BatchNorm. So, we use these two methods along with BP and compare them to CBP on semi-stationary problems. As we study online problems and we consider incremental updates, we used OnlineNorm (Chiley et al., 2019), an online variant of BatchNorm.

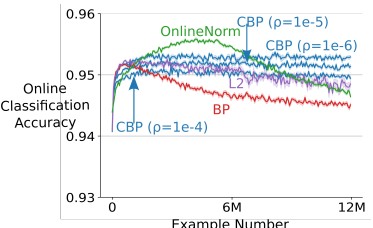

Figure 7: The online classification accuracy of various algorithms on Permuted MNIST. The performance of all algorithms except CBP degrade over time. CBP maintains a good level of performance for a wide range of replacement rates, $\rho$.

Figure 6 shows the online squared error for various learning algorithms with tanh and Relu networks. The online error is binned into bins of size 20k. The results are averaged over 100 runs. For all configurations of parameters for all algorithms, we used the same data: they had the same sequences of input-output pairs. All algorithms used SGD with step-size 0.01. For CBP and BP+L2, we used replacement-rate and weight-decay values that had the least total error over 1M examples. Figure 6 also shows the sensitivity of CBP and BP+L2 to replacement-rate and weight-decay respectively. OnlineNorm has two main parameters; they are used for computing running averages. The default values of these parameters are 0.99 and 0.999. We chose the best set of parameters from $\{0.9, 0.99, 0.999, 0.9999\} * \{0.9, 0.99, 0.999, 0.9999\}$. In Appendix A, we show the performance of these algorithms with other activation functions. In Appendix B, we also show the performance of these algorithms using Adam. For all the experiments with CBP in this paper, we use $\eta = 0.99$. And, for all experiments where we use CBP with SGD, we used $m = 100$.

Figure 6 shows that all three algorithms: CBP, BP+L2, and BP+OnlineNorm, perform significantly better than BP. However, BP+OnlineNorm is not stable for any of the activations as its performance

degrades over time. In comparison, BP+L2 regularization is stable for the tanh network but not the ReLU network. The parameter-sensitivity plot in Figure 6 shows that CBP and BP+L2 perform better than BP for most values of replacement-rate and weight-decay. And that CBP is much less sensitive to the parameter values than L2. For ReLU, CBP significantly outperformed BP+L2.

In Appendix C, we do an ablation study for different parts of our utility measure and compare it to other baselines like random utility and weight magnitude based utility (Mahmood and Sutton, 2013). We find that all the components of our utility measure are needed for best performance.

Figure 7 shows the performance of various learning algorithms on Permuted MNIST. We used SGD with a step size of 0.003. We binned the online accuracy into bins of size 60,000. The results are averaged over ten runs. And, for OnlineNorm, we did a parameter sweep for $\{0.9, 0.99, 0.999, 0.9999\} * \{0.9, 0.99, 0.999, 0.9999\}$. For L2, we chose the best weight-decay among $\{1e-3, 1e-4, 1e-5, 1e-6\}$.

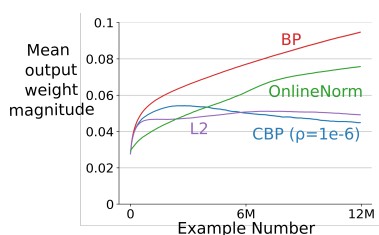

Figure 7 show that among all the algorithms, only CBP is stable as its performance does not degrade over time. The performance of all the other algorithms degrades over time.

Figure 8 shows that the average weight magnitude in the last layer increases over time for BP. Large weight magnitudes are problematic for both SGD and Adam. For SGD, large weights lead to blown-up gradients for the inner layer, which leads to unstable learning and saturated features. On the other hand, Adam has an upper bound on the weight change in a single update. Due to this upper limit on the effective change in each weight, the relative change in the function is smaller when the weights are large, which means slower adaptation. Both Adam and SGD on their own are unsuitable for Continual learning.

Figure 8: Evolution of the mean magnitude of the weights in the last layer with various learning algorithms on Permuted MNIST. One of the benefits of initial distribution is small weight magnitude, however, the weight magnitude continues to increase for BP. L2 is able to stop the increase in the weight magnitude, but its performance still gets worse. This suggests that small weight magnitude is not the only important benefit of the initial distribution. Other factors like diversity of features could also be playing an important role.

**Continual Learning in non-stationary RL problems**

We used the PPO algorithm as described by Schaulman et al. (2017). We used separate networks for policy and value function, and both had two hidden layers with 256 hidden units. We used *tanh* activation as it performs the best with on-policy algorithms like PPO (Andrychowicz et al., 2020).

To use CBP with PPO, we use CBP instead of BP to update the networks' weights. Whenever the weights are updated using Adam, we also update them using generate-and-test. We call the resulting algorithm *Continual PPO*, we describe it in Appendix D.

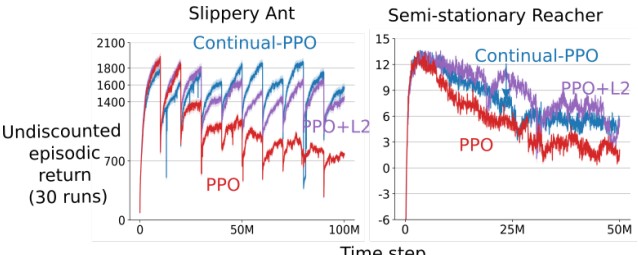

Figure 9 shows the performance of PPO, PPO+L2, and Continual PPO (CPPO) for Lecun initialization (Lecun et al., 1998) on our non-stationary RL problems. For L2, we chose the best weight-decay, and for CPPO we chose the best replacement-rate, maturity-threshold pair.

Figure 9: Performance of PPO, PPO+L2, and Continual-PPO on non-stationary RL problems. In both problems, the performance of PPO degraded rapidly. In contrast, the performance of Continual-PPO and PPO+L2 either degraded slowly or remained at a high level.

All other parameters were the same in PPO, PPO+L2, and CPPO. We describe the parameters in Appendix D. In the figure, the episodic return is binned among bins of 40,000 time-steps.

Figure 9 shows that the performance of PPO degrades a lot in both of our non-stationary RL problems. This degradation is similar to the degradation of Backprop's performance on semi-stationary problems, where it performed well initially, but its performance got worse over time. Figure 9 shows that both CPPO and PPO+L2 perform substantially better than PPO in both non-stationary

RL problems. On Slippery-ant CPPO performed best, where it continually performed as well as it did initially. While on semi-stationary Reacher, all algorithms fail.

## 5 RELATED WORKS

Continual learning or lifelong learning has been studied for a long time from several different perspectives. Ring (1998) considered it as a reinforcement learning problem in a non-Markovian environment. On the other hand, Caruana (1998) considered it as a sequence of tasks. There are two major goals in continual learning: memorizing useful information and adapting to the changes in the data stream changes by finding new useful information. Memorizing useful information is difficult for current deep-learning methods as they tend to forget previously learned information (McCloskey and Cohen, 1989; French, 1997; Parisi et al.,2019). Various ingenious methods have been proposed to tackle the forgetting issue (Kirkpatrick et al., 2017; Aljundi et al., 2018; Riemer et al., 2019; Javed et al., 2019); however, it still remains an unsolved problem. In this work, we focused on continually finding useful information but not on remembering useful information.

Multiple recent works have focused on adapting to the changes in the data stream by finding new useful information (Rusu et al. 2016; Finn et al., 2017; Wang et al., 2017; Nagabandi et al., 2019; Finn et al., 2019). However, none of the prior works studied problems with thousands of non-stationarities. The problem settings in most works were offline as they had two separate phases, one for learning and the other for evaluation. And in the evaluation phase, there were only a handful of non-stationarities. On the other hand, Finn et al. (2019) studied the fully online problem setting. Although this works investigated the interesting, fully online case, there were still a small number of non-stationarities. Also, the algorithms proposed by Finn et al. (2019) are not scalable as they require storing all the data seen so far. The closest work to ours is by Rahman (2021), where they studied learning in supervised learning problems where the target function changes over time.

Prior works on the importance of initialization have focused on finding the correct weight magnitude to initialize the weights. Glorot et al. (2010) showed that it is essential to initialize the weights so that the gradients do not become exponentially small in the initial layers of a network with sigmoid activations. Sutskever et al. (2013) showed that initialization with small weights is critical for sigmoid activations as they may saturate if the weights are too large. More recent work (He et al., 2015) has gone into ensuring that the input signal's scale of magnitude is preserved across the layers in a deep network. Despite all these works on the importance of initialization, the fact that its benefits are only present in the beginning but not continually has been overlooked as most of these works focused on offline learning, where learning has to be done just once.

## 6 CONCLUSION AND LIMITATIONS

In this work, we studied learning problems that have a large number of non-stationarities. We showed that in these problems, the performance of Backprop degrades over time, and it loses its ability to adapt. In other words, we showed that neural networks have a "decaying plasticity" problem. Ours is the first result showing this degradation in Backprop's ability to adapt. This result means that Backprop can be unsuitable for continual adaptation.

We proposed the Continual Backprop (CBP) algorithm to get a robust continual learning algorithm where both components of Backprop are present continually. CBP extends the benefits of special random initialization throughout learning using a generate-and-test process. Our generate-and-test algorithm is the first one that is compatible with modern deep learning. We showed that CBP could continually adapt even after thousands of non-stationarities. Additionally, we showed that Backprop with L2 regularization is also able to adapt continually in some problems, but it fails in many others.

The main limitation of our work is that our tester is based on heuristics. Even though it performs well, future work on more principled testers will improve the foundations of Continual Backprop. Continual learning is becoming more and more relevant in machine learning. These systems can not use Backprop or its variants in their current form, as they are not stable in non-stationary problems. We expect that by establishing Backprop's degrading adaptivity, our work will invite many solutions similar to how many works have addressed "catastrophic forgetting". We foresee that our work will open a path for better Continual Backprop algorithms.

**Ethics Statement:** In this work, we did fundamental research in continual learning. In the long term, our work may help develop continually learn systems for the real world. Continual learning can enable learning systems that scale with data and experience under constant memory requirements. This scaling will make learning systems accessible to a large part of the population, who lack easy access to massive computation, which is essential for current AI systems. Continual learning can also enable learning in remote locations where current expensive methods face difficulty due to limited on-site connectivity and memory resource. As for any potential negative impacts, they are not unique to our work, and instead, they will be shared with other works on continual learning.

**Reproducibility Statement:** To ensure the Reproducibility of the experiments in this paper, we will make the source code for all the algorithms and environments publicly available after acceptance. To ensure that our work is fully accessible to the public, we used a publicly available dataset (MNIST) and simulator (PyBullet) in our experiments. Finally, to replicate the experiments, we have given the details of all the hyperparameters in the main text and Appendix D.

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

APPENDIX

## A    MORE EXPERIMENTS WITH BACKPROP ON THE BIT-FLIPPING PROBLEM

In Figure 3 we showed that the performance of Backprop for ReLU and tanh activations gets worse over time. In Figure 10, we show that the performance of Backprop also degrades for other activation functions.

In Figure 3 and 10 we ran the experiment for 1M examples. For this length of experiment it may seem that the error for smaller step-size of 0.001 reduced or stayed the same for some activation functions. In Figure 11, we show that even for a step-size of 0.001, if we let Backprop track for longer (5M), the error for all activations increases significantly or stays at a high level. Again, in Figure 11, we bin the online prediction error into bins of size 20,000 and the shaded region in the figure shows the standard error of the binned error.

Next, we evaluate the performance of BP when using Adam instead of SGD. In all the experiments, we used Adam with betas (0.9, 0.999). First, in Figure 12, we show that the performance of BP gets worse over time for a wide range of step-sizes for all activation functions. Then in Figure 13, we show that even for a smaller step-size of 0.001, after tracking for longer (5M examples), the error for all activations increases significantly.

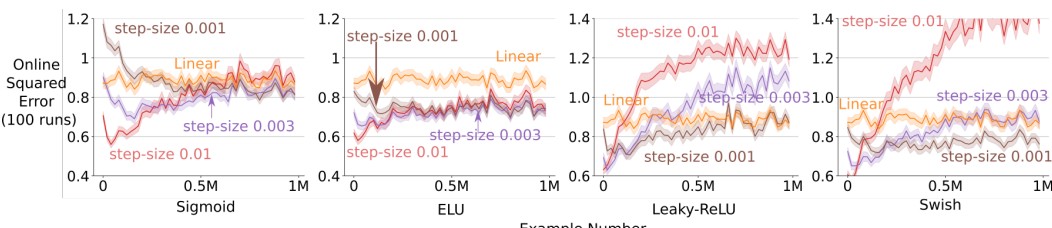

Figure 10: Learning curve of BP on the Bit-Flipping problem for various activation functions using SGD. After performing well initially, the performance of Backprop gets worse over time for all activation functions.

## B    MORE EXPERIMENTS WITH CBP ON THE BIT-FLIPPING PROBLEM

First, we look at the performance of CBP with the remaining activation functions to complement the results shown in Figure 6. The first half of Figure 14 shows the online performance of various algorithms, and the second half shows the average loss over the experiment's full length (1M examples) for various values of hyperparameter replacement-rate (for CBP) and weight-decay (for BP+L2).

To properly use Adam with CBP, we need to modify Adam. Adam maintains estimates of the average gradient and the average squared gradient. CBP resets some of the weights whenever a feature is replaced. Whenever CBP resets a weight, we set its estimated gradient and squared gradient to zero. Adam also maintains a 'timestep' parameter to get an unbiased estimate of the average gradient. Again, whenever CBP resets a weight, we set its timestep parameter to zero.

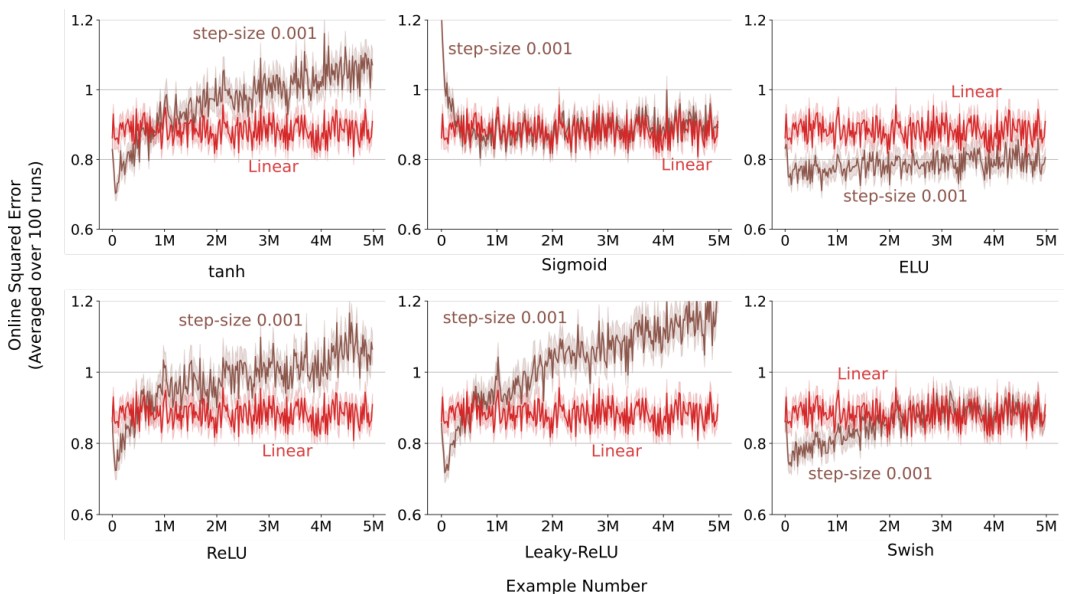

Figure 11: SGD with a smaller step-size on the Bit-Flipping problem. After running for 1M steps in Figure 3, it may seem that performance for step-size of 0.001 does not worsen over time. However, after running the experiment for longer, 5M steps, we found that the error for step-size of 0.001 either increases significantly or remains significantly higher than larger step-sizes.

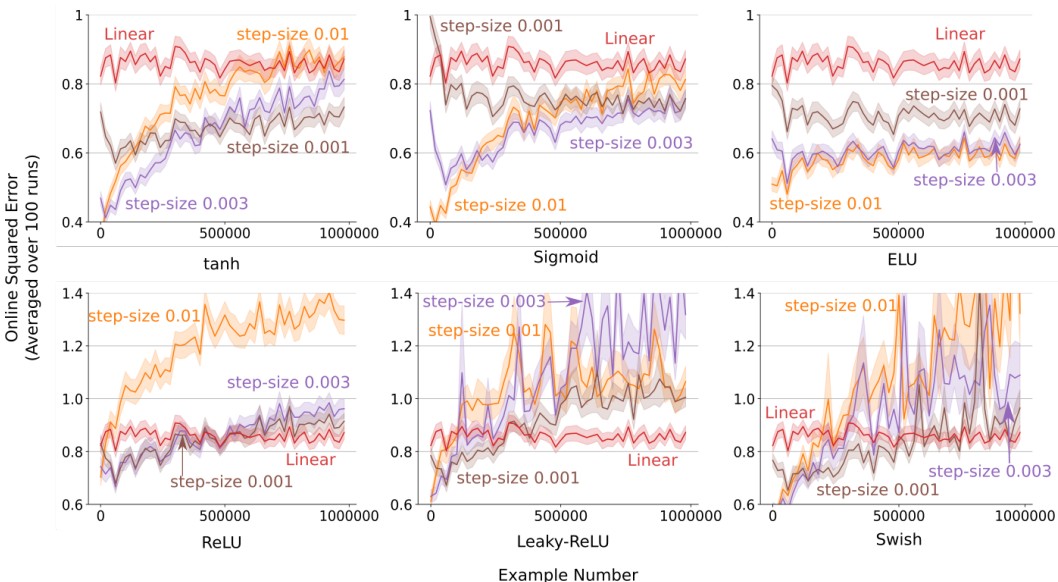

Figure 12: The learning curve on the Bit-Flipping problem using Adam. Again, after performing well initially, the error goes up for all activation function. Backprop's ability to track becomes worse under extended tracking even with Adam.

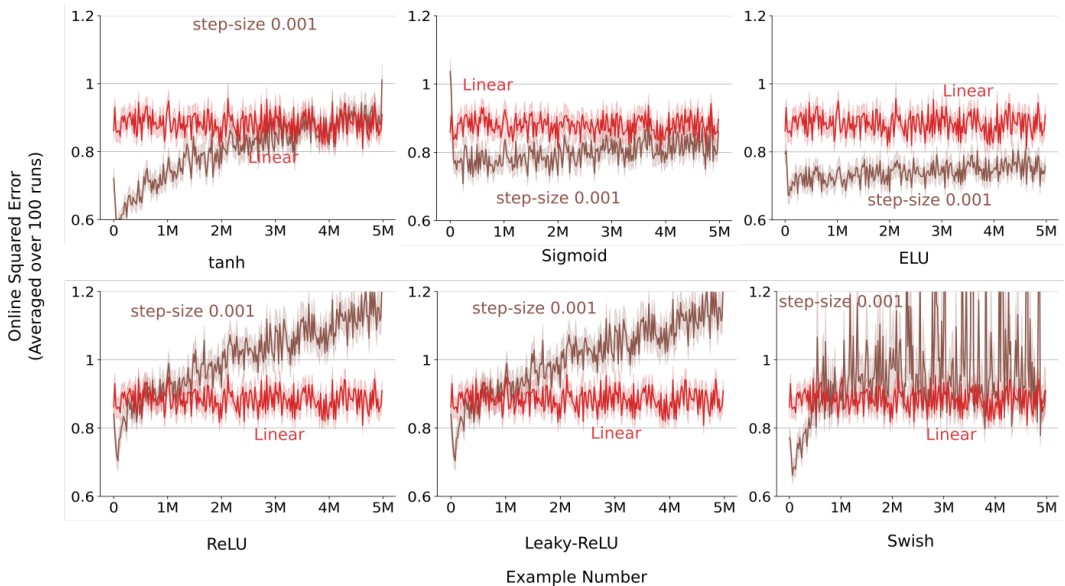

Figure 13: Adam with a smaller step-size on the Bit-Flipping problem. Like SGD, after running for 1M steps in Figure 11, it may seem that the performance for step-size of 0.001 does not worsen over time. However, after tracking for 5M examples, the error for step-size of 0.001 increase too.

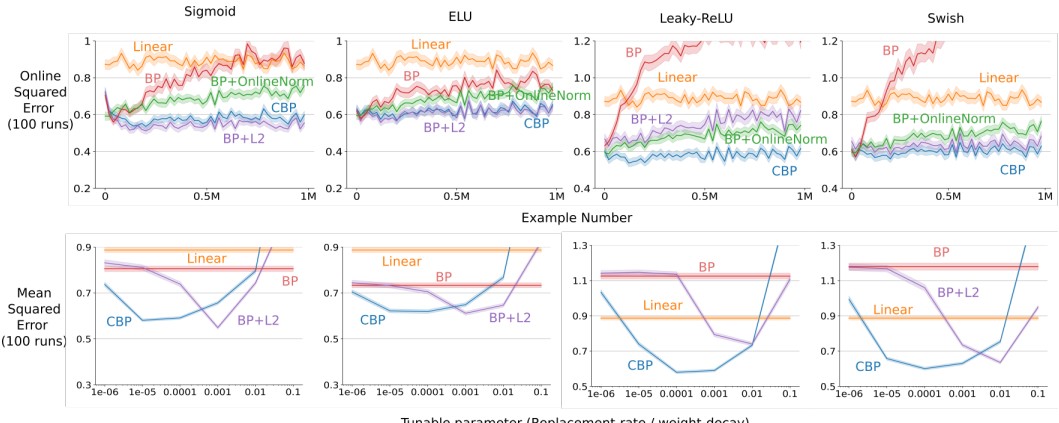

Figure 14: Parameter sweep and online performance of various learning algorithm using SGD for step-size of 0.01 on the Bit-Flipping problem. Only CBP and BP+L2 are suitable for continual learning, the performance of all the other algorithms degrade over time. For all activation functions, CBP and BP+L2 perform significantly better than BP for almost all values of hyperprameters. CBP is less sensitive to its hyperparameter than BP+L2.

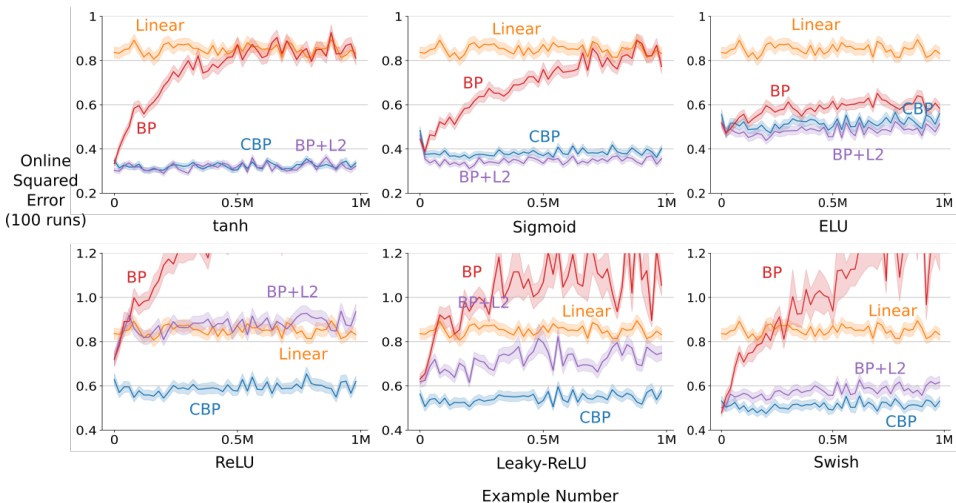

Figure 15: The learning curves of Backprop(BP), Backprop with L2(BP+L2) and Continual Backprop (CBP) on the Bit-Flipping problem using Adam with step-size of 0.01. Both BP+L2 and CBP are performed robustly on this problem as their performance did not degrade over time. The best parameters for CBP and BP+L2 are chosen separately for each activation. For all activations, CBP is either significantly better than BP+L2 or just slightly worse.

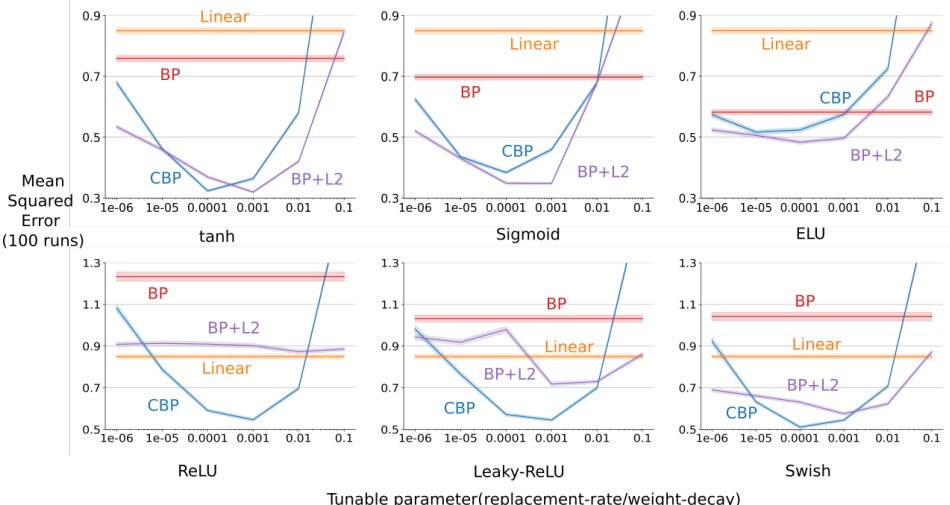

Figure 16: Parameter sweep for CBP and BP+L2 when used with Adam for step-size of 0.01 on the Bit-Flipping problem. Again, for all activation functions both CBP and L2 perform significantly better than BP for most values of replacement-rate and weight-decay. For some activations like Sigmoid and ELU, BP+L2 performs better than CBP, while for other activations like ReLU and Leaky-ReLU, CBP performs better than BP+L2.

---

**Algorithm 2:** Continual Backprop (CBP) with Adam for a feed forward neural network with $L$ hidden layers

---

**Set:** step-size $\alpha$, replacement rate $\rho$, decay rate $\eta$, and maturity threshold $m$ (e.g. $10^{-4}$, $10^{-4}$, 0.99, and 1000)
**Initialize:** Set moment estimates $\beta_1$, and $\beta_2$ (e.g., 0.9, and 0.99)
**Initialize:** Randomly initialize the weights $\mathbf{w}_0, ..., \mathbf{w}_L$. Let, $\mathbf{w}_l$ be sampled from a distribution $d_l$
**Initialize:** Utilities $\mathbf{u}_1, ..., \mathbf{u}_L$, average feature activation $\mathbf{f}_1, ..., \mathbf{f}_l$, and ages $\mathbf{a}_1, ..., \mathbf{a}_L$ to 0
**\* Initialize:** Moment vectors $\mathbf{m}_1, ..., \mathbf{m}_L$, and $\mathbf{v}_1, ..., \mathbf{v}_L$ to 0, where $\mathbf{m}_l$, $\mathbf{v}_l$ are the first and second moment vectors for weights in layer $l$
**\* Initialize:** timestep vectors $\mathbf{t}_1, ..., \mathbf{t}_L$ to 0, where $\mathbf{t}_l$ is the timestep vector for weights in layer $l$
**for** *each input $x_t$* **do**
> **Forward pass:** pass input through the network, get the prediction, $\hat{y}_t$
> **Evaluate:** Receive loss $l(x_t, \hat{y}_t)$
> **Backward pass:** update the weights using stochastic gradient descent
> **\* Update timestep:** Increase $\mathbf{t}_1, ..., \mathbf{t}_L$ by 1
> **\* Update moment estimates:** Update $\mathbf{m}_1, ..., \mathbf{m}_L$, and $\mathbf{v}_1, ..., \mathbf{v}_L$, using the newly computed gradients
> **for** *layer $l$ in $1 : L$* **do**
> > **Update age:** $\mathbf{a}_l += 1$
> > **Update feature utility:** Using Equation 5
> > **Find eligible features:** Features with age more than $m$
> > **Features to replace:** $n_l * \rho$ of eligible features with smallest utility, let their indices be $\mathbf{r}$
> > **Initialize input weights:** Reset input weights $\mathbf{w}_{l-1}[:, \mathbf{r}]$ using random samples from $d_l$
> > **Initialize output weights:** Set $\mathbf{w}_l[\mathbf{r}, :]$ to zero
> > **Initialize utility, feature activation, and age:** Set $\mathbf{u}_{l,\mathbf{r},t}$, $\mathbf{f}_{l,\mathbf{r},t}$, and $\mathbf{a}_{l,\mathbf{r},t}$ to 0
> > **\* Initialize moment estimates:** Set $\mathbf{m}_{l-1}[:, \mathbf{r}]$, $\mathbf{m}_l[\mathbf{r}, :]$, $\mathbf{v}_{l-1}[:, \mathbf{r}]$, and $\mathbf{v}_l[\mathbf{r}, :]$ to 0
> > **\* Initialize timestep:** Set $\mathbf{t}_{l-1}[:, \mathbf{r}]$, and $\mathbf{t}_l[\mathbf{r}, :]$ to 0

---

\* These are Adam specific updates in CBP
The inner for-loop specifies generate-and-test based updates

---

In Figures 15 and 16, we show that CBP with Adam performs significantly better than BP with Adam for a wide range of replacement-rates.

## C  ABLATION STUDY FOR THE UTILITY MEASURE

The final utility measure that we used in this work consists of two parts, the contribution-utility and the adaptation-utility. Now, we will compare various parts of the utility measure on the Bit-Flipping problem. We use a learning network with tanh activation, and we use the Adam optimizer with a step-size of 0.01. Additionally, we also compare our utility measure with random utility and weight-magnitude-based utility (Mahmood and Sutton, 2013). The results for this comparison are presented in Figure 17a We compare the following utility measures.

- Random utility: Utility, $r$ at every time-step is uniformly randomly sampled from $U[0, 1]$
$$r_{l,i,t} = rand(0, 1)$$

- Weight-magnitude based utility: $wm$ at every time-step is updated as:
$$wm_{l,i,t} = (1 - \eta) * \sum_{k=1}^{n_{l+1}} |w_{l,i,k,t}| + \eta * wm_{l,i,t-1}$$

- Contribution utility, $c$, at every time-step is updated as described in Equation 1

- Mean corrected contribution utility, $z$, at every time-step is updated as described in Equation 2

- Adaptation utility, $a$, at every time-step is updated as:
$$a_{l,i,t} = (1 - \eta) * \frac{1}{\sum_{j=1}^{n_{l-1}} |w_{l,j,i,t}|} + \eta * a_{l,i,t-1}$$

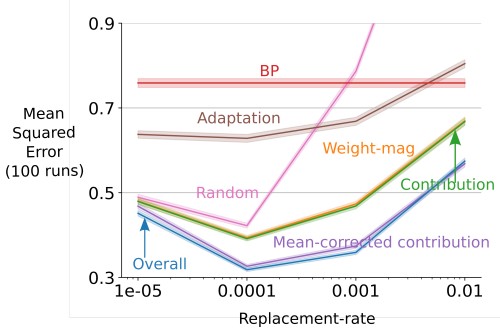 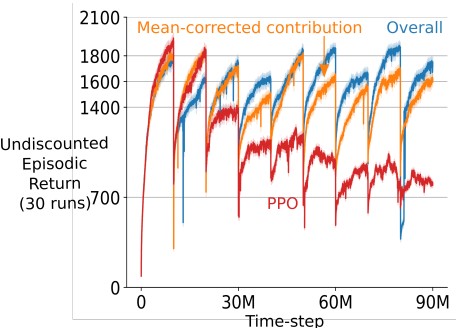

(a) Parameter sweep for various utility measures on the Bit-Flipping problem. The overall utility measure performs better than all other measures of utility including the weight-magnitude based utility (Mahmood and Sutton, 2013) and random utility.

(b) Continual PPO with overall utility and mean-corrected contribution utility on Slippery Ant. Both utility measures performs a lot better than PPO and overall utility perform better than mean-corrected contribution utility.

Figure 17: CBP with different utility measures on different problems

- Overall utility, $u$ ,at every time-step is updated as described in Equation 5

Next we compared the two best performing utility measures, overall and mean-corrected contribution, on Slippery Ant. The results are presented in Figure 17b. The results show that both utility measures perform significantly better than PPO. But, the overall utility performs significantly better than the other utility measure. This difference in more pronounced near the end, when Continual PPO with the overall utility measure performs almost as well as it did at the beginning.

# D    CONTINUAL PPO

We used the following values of the parameters for PPO, PPO+L2 and CPPO in our experiments on non-stationary RL problems. For CPPO and PPO+L2 specific parameters, we chose the best performing values.

- Policy Network: (256, tanh, 256, tanh, Linear) + Standard Deviation variable

- Value Network (256, tanh, 256, tanh, linear)

- iteration size: 4096

- num epochs: 10

- mini-batch size: 128

- GAE, $\lambda$: 0.95

- Discount factor, $\gamma$: 0.99

- clip parameter: 0.2

- Optimizer: Adam

- Optimizer step-size: $1e-4$

- Optimizer $\beta$s: (0.9, 0.999)

- weight-decay (for L2): $\{10^{-3}, 10^{-4}, 10^{-5}, 10^{-6}\}$

- replacement-rate, maturity-threshold (for CPPO): $\{(10^{-3}, 1e2), (10^{-3}, 5e2), (10^{-4}, 5e3), (10^{-4}, 5e2), (10^{-5}, 1e4), (10^{-5}, 5e4)\}$

---

**Algorithm 3:** Continual PPO

---

**Initialize:** All the required parameters for PPO
**Initialize:** All the required parameters for CBP with Adam in Algorithm 1 in the appendix
**for** *iteration = 1, 2 ...* **do**
    **Collect data:** Run the current policy to collect a set of trajectories
    **for** *epochs = 1, 2 ...* **do**
        Divide and shuffle the collected trajectories into mini-batches
        **for** *each mini-batch* **do**
            Compute the objectives for policy and value networks
            Update the weights of both networks using Adam
            Update the weights of both networks using generate-and-test

---

## E  COMPUTE USED

All the experiments in this work were performed on Intel Gold 6148 Skylake CPUs or NVIDIA Tesla P100 GPUs. On the Bit-Flipping problem, all the algorithms took around 30 minutes for a single run with 1M examples. The total compute used for all the experiments on the Bit-Flipping problem was around 12960 CPU-days or 36 CPU-years. On the Permuted-MNIST problem, each took about 8 GPU hours, which totals 110 GPU days for all experiments. Finally, the experiments for non-stationary RL tasks took up to 90hours for 100M time-steps, the total compute used for all experiments on the non-stationary RL problems was 1980 CPU-days or 5.4 CPU years.

## F  AFFECT OF THE SPEED OF THE DISTRIBUTION CHANGE ON DECAYING PLASTICITY

Figure 18 shows the online classification accuracy of BP on a deep ReLU-network with different speeds of distribution change. In the fastest changing distribution, the distribution changes after every 10k examples, while in the slowest changing distribution, it changes after every 1M examples. We used SGD with a step-size of 0.01. The network has three hidden layers with 2000 units each.

The Figure 18 shows that the drop in performance is fastest for the fastest changing distribution.

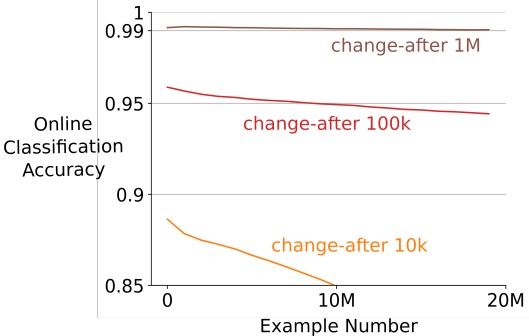

Figure 18: The online classification accuracy of a deep ReLU-network on Permuted MNIST with different speeds of distribution change. The time after which the distribution changes varies from 10k to 1M. The online accuracy is binned among bins of size 1M. It shows that the speed of the decay in performance increases with the increase in the speed of distribution change.

## G  A DEEPER LOOK IN THE NETWORKS: FEATURE SATURATION AND VANISHING GRADIENTS

Figure 19a shows the number of saturated features for tanh-network on the Bit-Flipping problem. We call a feature is saturated when the magnitude of its output is more than 0.9. Figure 19b shows the average magnitude of gradients for the weights in the input layer. For different algorithms, the same values of hyper-parameters are used as we did in Figure 6. In both Figure 19a and Figure 19b, the data is binned into bins of 20,000 examples.

Figure 19a shows that an increasingly large fraction of features saturate over time for Backprop. When a feature gets saturated, the network loses some of its representation power which causes worse performance. With Backprop, almost 90% of the features are saturated. Figure 19b shows that with BP, the magnitude of the gradient of the input weights becomes significantly smaller after

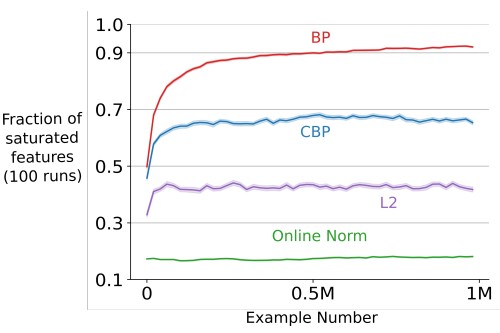
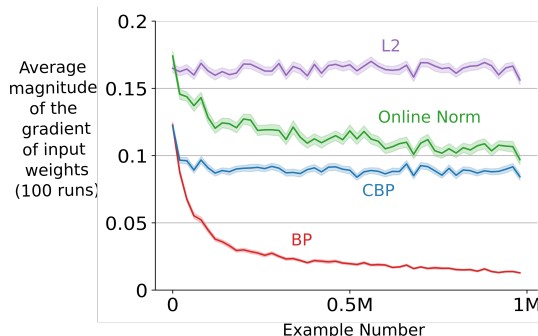

(a) Fraction of saturated features for different algorithms. Despite having the least number of saturated features, Online Norm doesn't perform well. This means that feature saturation is not the root cause of decaying plasticity.

(b) The average magnitude of the gradient for the input weights for different algorithms. Again, despite having a large gradient for the input weights, Online Norm doesn't perform well. This means that the decreasing gradient of the input weights is not the full explanation of the decaying plasticity of BP.

Figure 19: Evolution of feature saturation and the gradient of the input weights for various learning algorithms.

some time. A small gradient means that the weights change less, and the network loses its ability to adapt. This increase in saturated features and the small gradient of the input weights partially explain why BP performs worse after extended tracking. However, the evolution of the level of feature saturation does not correlate with the performance of all algorithms. Figure 6 shows that CBP and L2 have the least online error, but the online error with of OnlineNorm is much larger than CBP or L2. However, in Figure 19a, OnlineNorm has the least number of saturated features. Similarly, in Figure 19b, OnlineNorm has a higher input gradient than CBP, but its performance is worse than that of CBP. This means that neither feature saturation nor the magnitude of input gradients are fully explaining the behavior of all algorithms.

## H    AFFECT OF NETWORK SIZE ON DECAYING PLASTICITY

Figure 20 shows the online classification accuracy of BP with different deep ReLU-networks. Both networks had three hidden layers. The smaller network had just 100 hidden units, while the wider network had 1000 hidden units. We used SGD with a step-size of 0.003. Figure 20 shows that the drop in performance is fastest for the smaller network.

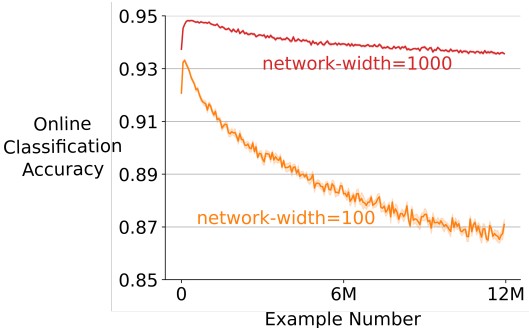

Figure 20: The online classification accuracy of deep ReLU-networks with different widths on Permuted MNIST The distribution changes after every 60k examples, and the online accuracy is binned among bins of size 60k. It shows that the speed of the decay in performance is larger for smaller network.

