# OpenReview forum: "Continual Backprop: Stochastic Gradient Descent with Persistent Randomness"
_ICLR.cc/2022/Conference — ICLR 2022 Submitted_

### Official Review · Reviewer_Evj3 · 2021-10-31

**Correctness:** 4
**Technical Novelty And Significance:** 3
**Empirical Novelty And Significance:** 3
**Recommendation:** 5
**Confidence:** 4

**Main Review:**

Strong Points

* The paper takes one of the most import issues in continual learning: non-stationary online CL setting. For me, the problem itself is real and practical.

* The proposed approach is reasonable and addresses the problem of weight randomnization in CL setting.

* Overall, the paper is well written. In particular, the Related Work section has a nice flow and puts the proposed method into context. Despite the method having limited novelty, the method has been well motivated by pointing out the limitations in SOTA methods.

Weak Points

* The proposed generate-and-test algorithm is built upon previous work by Mahmood and Sutton 2013. So the novelty is limited.

* Although the paper focuses on fast adaptation to new tasks in CL. It is important to evaluate how the proposed method deals with catastrophic forgetting. I would like to see experimental results on the catastrophic forgetting evaluation.

* "Continual backprop" in the title and paper does not reflect the attributes of the proposed algorithm. I would suggest changing "Continual backprop" to "Continual reinitialization" as a more appropriate summary of the paper and the algorithm.

* "feature" in the paper is used to refer to a hidden unit in a network (See page 5). However, "feature" commonly refers to a vector extracted from an embedding layer in a network. I would suggest that the authors change "feature" to "hidden unit" in order to avoid misunderstanding.

* Fig. 9, in semi-stationary reacher setting, PPO+L2 performs better than the proposed continual PPO. It's worth discussion the reasons for the results.



**Summary Of The Paper:**

This paper investigates the problem of fast adaptation in a non-stationary online continual learning(CL) setting. It argues that keeping weight randomnization is important to fast adaptation in CL. However, current CL methods only performs weight randomization in the beginning of the algorithm; the weights loss randomness overtime, leading to degraded model performance. The paper presents a continual weight reinitialization algorithm to overcome the issue. In particular, it proposes to evaluate the utility of each hidden unit -- including importance to the current task and adaptation capability. Then selects a set of hidden units with low score and resets their incoming and outgoing weights. The authors conduct experiments to evaluate the performance of the proposed method.



**Summary Of The Review:**

Overall, I vote for marginally accepting. I like the idea of continual reinitialization and handling it by the proposed generate-and-test method. My major concern is about the limited novelty of the paper -- its build upon previous work by Mahmood and Sutton 2013, and some misleading terms such as "continual backprop" and "feature".
Hopefully the authors can address my concern in the rebuttal period.

[After rebuttal]:
After reading the other reviewer's comments and authors' feedback, I would like to lower my score due to the same concerns as other reviewers as mentioned.

---

> ### Author Response · Authors · 2021-11-12
> **We address the weak points**
>
> Thank you for your time and valuable feedback. We are glad to see that you appreciate the importance of non-stationary online CL.
>
> We think that the main novelty of our paper is to establish the existence of “decaying plasticity,” a previously unknown phenomenon.
>
> We agree that it is important to study how methods designed to deal with “catastrophic forgetting” and “decaying plasticity” affect the other phenomenon. However, that work is beyond the scope of our current work. We are focused on establishing the existence of decaying plasticity and presenting algorithms that can overcome it. A future work that studies how various continual learning algorithms address catastrophic forgetting and decaying plasticity will be very important to the field.
>
> We will change the word “feature” to “hidden unit” in order to avoid confusion.
> The result in Figure 9 on the semi-stationary reacher is interesting. However, we do not fully understand why PPO+L2 is working better in that problem. That’s why we have not included a discussion for that result. It will be an exciting direction for future work to find out under which conditions does L2 performs better than CBP.

---

### Official Review · Reviewer_kzUn · 2021-11-02

**Correctness:** 3
**Technical Novelty And Significance:** 2
**Empirical Novelty And Significance:** 2
**Recommendation:** 3
**Confidence:** 4

**Main Review:**

STRENGTH

The topic of learning online, in continual, incremental or any other online setting, is certainly of high and growing interest in the community. Works trying to tackle the problem in its fundamental aspects, starting from the weight adjustment algorithms are very much interesting for the community. In this respect, this paper certainly takes an interesting and somewhat original view into the difficulties of learning online.

Additionally, the empirical analysis covering both supervised learning and reinforcement learning scenarios is certainly a strong aspect of the paper, at least for what pertains breadth of application (though, as discussed in the following, it comes at the cost of depth of the results).

WEAKNESSES

The paper has some substantial issues, ranging from solidity of the claims, scholarly value, novelty as well as empirical validation of the paper ideas.

In terms of claims, I strongly disagree with the sentence in the concluding paragraph of the introduction about “…we contribute to the understanding of why BP and its variants fail in non-stationary problems”.  The paper does not help understanding the root causes of the problem. It  provides empirical results showing that a certain amount of randomness reintroduced dynamically into the model helps in keeping the model able to learn. This is far from explaining why this happens. For instance, I would have expected to see at least experiments to rule-out the fact that the inability to learn is due to gradient vanish (this can be easily plotted by monitoring the gradient magnitude). I had the feeling that the departure from the small weight initialization might cause reduced learning due to larger weights bringing the neurons into the saturated parts of their activation functions. My first intuition was to think about regularizing on the norm of the weights, which is in fact providing effective results in the empirical analysis. Apart from the permuted MNIST case, the L2 regularization seems to be providing nearly the same advantage as the proposed CBP algorithm.

In terms of scholarly value, the paper is often confusing with respect to the community it is referring to: it is somehow linking the paper both to the continual learning community as well as to the community of learning in non-stationary environments. The paper needs to make a clear choice as the underlying assumptions (e.g. forgetting Vs remembering), the related literature and the related benchmarks differ substantially. Currently, the paper is in between the two worlds, and unsatisfactory in the way it addresses the point above with respect to both. For instance, in the continual learning community, one might wonder on what are the relationships between the proposed method and other weight-constraining/regularization approaches such as EWC and LWF.

In terms of novelty, the solution proposed in the paper builds heavily on existing approaches, mostly on generate-and-test extended on non-major aspects, such as support for multi-layer perceptrons and non-LTU activations. Hence, I do not see much original content in the proposed CBP algorithm, whereas there might be some truly original insights coming from the intuition of the root causes of BP inability in learning online (but as said, these are underdeveloped in the paper).

The empirical analysis, as anticipated, could make good use of experiments showing the behaviour of the gradient magnitude across learning iterations. Current experiments, although commendable in the fact that they cover both supervised and reinforcement learning, do not provide substantial proof of CBP being better than existing weight regularization techniques.


**Summary Of The Paper:**

The paper discusses a modification of the BP algorithm, the Continual BP (CBP), that founds on the intuition that  standard BP cannot work in online learning scenarios due to weights departing from the initial condition of being small and random, after some training experience. The paper contributes with an empirical analysis of the phenomenon as well an algorithm to address the issue.

**Summary Of The Review:**

The topic of the paper is of potential interest for the community and the empirical analysis sufficiently broad. On the negative end, the depth of technical discussion and of the empirical analysis does not support the claims. Originality of the contributed CBP algorithm is minor.

---

> ### Author Response · Authors · 2021-11-13
> **We address the concerns raised by the reviewer.**
>
> Thank you for your time and review. We will address your points one by one.
>
> About solidity of claims
>
> Note that we never claimed to “... help understanding the root causes of the problem”. We said that “…we contribute to the understanding of why BP and its variants fail in non-stationary problems”. We claim that we contribute to the understanding for the following reason:
> 1. It is well known in the deepRL community that vanilla BP is not well-suited for deep RL. People make concessions like target value networks to try to overcome this obstacle. The RL problem is non-stationary because the input distribution changes as the agent’s behaviour changes and the target function changes because the value function continually changes. However, the reasons for the failure of vanilla backprop in these non-stationary problems are not known.
> 2. People guess that the failure of BP in deep RL is because of catastrophic forgetting.
> 3. However, in this work, we showed that vanilla BP also displays “decaying plasticity.”
> 4. Thus, we have shown that aside from forgetting, “decaying plasticity” is another phenomenon that affects BP in non-stationary problems. Hence, “contributing to the understanding”, by showing another failure mode of BP in non-stationary problems.
>
> Thank you for pointing out feature saturation and low gradients in the input layers. We originally found that features saturate and gradients vanish in BP. That was the motivation for using L2 and OnlineNorm. However, we did not include those results in the original submission because it does not explain the performance of all algorithms. We have added those results in a new section in the appendix. Please have a look.
>
>
> Scholarly Value:
>
> We disagree that the “continual learning community” and the “community of learning in non-stationary environments” are different. If you want to force a separation, you can say that the “community of learning in non-stationary environments” is a subset of the “continual learning community.” We think the confusion is happening because you narrowly define the “continual learning community” as the group of people working on the catastrophic forgetting problem. [1] said that “The ability to continually learn over time by accommodating new knowledge while retaining previously learned experiences is referred to as continual or lifelong learning.” According to this definition, “The ability to continually learn” is an essential part of continual learning. In our paper, we are showing that BP loses its “ability to continually learn.” So, our work is part of the continual learning literature even though we are not working on the catastrophic forgetting problem.
>
> As for your comment that “the paper needs to make a clear choice about the underlying assumptions,” We have clearly mentioned multiple times in the paper that “we do not study the forgetting problem” (page-2) and that “In this work, we focused on continually finding useful information but not on remembering useful information” (page-9).
>
>
> Novelty:
>
>
> Yes, our work builds upon prior work on generate-and-test. Maybe you can argue that extending generate-and-test to deep networks is not novel enough, and so our paper lacks methodological novelty. But, you can not argue that paper as a whole lacks novelty because we establish the existence of “decaying plasticity,” a previously unknown phenomenon.  We urge the reviewer to judge the merit of the work based on this discovery. Do you think the discovery of “decaying plasticity” is important and novel?
>
> Empirical validation:
> Please have a look at our new section in the appendix, where we plot the behaviour of the gradient magnitude across learning iterations.
>
>
> [1] Parisi, G. I., Kemker, R., Part, J. L., Kanan, C., & Wermter, S. (2019). Continual lifelong learning with neural networks: A review. Neural Networks, 113, 54-71.

---

> > ### Comment · Reviewer_kzUn · 2021-11-23
> > **Thanks for your response**
> >
> > Let me begin by saying that I have deeply appreciated that the Authors promptly responded with additional experiments deepening the insights on what I believe is the most interesting and novel aspect of the paper, i.e. understanding BP in online settings.
> >
> > This said, I will go on by answering to the latest question in the rebuttal: "Do you think the discovery of “decaying plasticity” is important and novel?" I do think it could be important, possibly novel, but unfortunately the paper is not there yet. Again, I do have a different understanding of what it means "contribute to the understanding of why BP and its variants fail in non-stationary problems": I do know that the authors with this sentence never meant to say that the paper "help understanding the root causes of the problem”, but to me that's what a paper claiming novelty and impact on the topic should do. Should provide at least hints and insights on the causes of a phenomenon which is only measured empirically, on quite artificial setting and in restricted architectures. The point of the experiments is often that:
> > - Provided that online learning is ran for enough time (maybe 1M steps, with smaller learning rates maybe 5M, with even smaller rates maybe 10M, and so on)
> > - Provided that we do not use regularization/normalization approaches, or that such regularization approaches fail in some cases, for which it is not yet clear a pattern
> > then it is observed something that is defined decaying plasticity and it is a general issue of BP. Then I am afraid the empirical evidence in the paper is not enough convincing to make up for the lack of a discussion on root causes. There is simply not enought insight into the why and the conditions favouring it.
> >
> > A final point about continual learning: whetherwe like it or not, communities tend to define their perimeter. There is a clearly defined perimeter in the community that recognizes under the umbrella term of continual learning. And for that community Catastrophic forgetting is central. A community also has benchmarks and empirical setups. Positioning a paper into a community with a certain terminology creates expectations as what the paper is expected to show, as all reviews have highlighted. While decaying plasticity can be an orthogonal theme to catastrophic forgetting, a paper directed towards the continual learning community that does not consider the latter is likely to achieve little impact, in my opinion.

---

> > > ### Author Response · Authors · 2021-11-25
> > > **Thank you for your reply**
> > >
> > > Thank you for your reply, we appreciate that you clarified your position.
> > > We will start by addressing your different points that the empirical evidence is not enough.
> > > * Most of the settings in the paper are not artificial, and the architectures are not restricted. The permuted MNIST problem is not artificial, and the RL problems (Ant and Reacher) are part of the OpenAI Gym, the standard benchmark in RL. The network we used on Permuted MNIST is around 30 times larger than a typical feed-forward network used on MNIST [1]. Moreover, the networks we used on RL problems are the same size as standard networks used on these problems [2].
> > > * The experimental condition that “Provided that online learning is ran long enough” is precisely what a continual/lifelong learning system should do; it should be able to learn forever. However, as you point out, our experiments show that eventually, the performance of BP gets worse; it just takes different times for different step-sizes.
> > > * Thirdly, we will clarify that when we say “contribute to the understanding of why BP and its variants fail in non-stationary problems,” we refer to Adam when we talk about the variant, not BP+L2 or Online Normalization. A more precise version of that statement will be “contribute to the understanding of why BP and its variant Adam fail in non-stationary problems.”
> > > * The experiments show that the performance of BP degrades for activation and step-sizes, for both Adam and SGD, in both Supervised and RL problems. Our new experiments also show that the performance degrades for a wide range of speeds of distribution change and various network sizes. These results conclusively show that decaying plasticity is a general phenomenon.
> > >
> > > Thanks for explaining the community point of view from your perspective. Although prior works on continual learning (CL), such as [3] published at ICLR 2021, discussed the importance of addressing other critical issues, including plasticity, in CL than catastrophic forgetting, it is understandable that yours can be a common perception in the community. Likewise, we will clarify that our work is not about catastrophic interference but another critical problem discovered in the current work.
> > >
> > > [1] Dettmers, T., & Zettlemoyer, L. (2019). Sparse networks from scratch: Faster training without losing performance. arXiv preprint arXiv:1907.04840.
> > > [2] Haarnoja, T., Zhou, A., Abbeel, P., & Levine, S. (2018, July). Soft actor-critic: Off-policy maximum entropy deep reinforcement learning with a stochastic actor. In International conference on machine learning (pp. 1861-1870). PMLR.
> > > [3] Veniat, T., Denoyer, L., & Ranzato, M. A. (2020). Efficient continual learning with modular networks and task-driven priors. arXiv preprint arXiv:2012.12631.

---

### Official Review · Reviewer_jf4G · 2021-11-06

**Correctness:** 3
**Technical Novelty And Significance:** 3
**Empirical Novelty And Significance:** 4
**Recommendation:** 6
**Confidence:** 4

**Main Review:**

Strengths:
- This paper proposes and tackles a novel important problem in continual learning that has not been observed in prior work.
- They demonstrate on both proposed solution is able to perform significantly better on this problem and does not degrade in performance over time compared to the vanilla back propagation on feed-forward networks on CV, RL, and bit-flipping problems
- They show empirical results on a variety of different Feed-forward networks with different hyperparameters and activation functions. They compare against

Weakness:
- The paper does not compare the tradeoff of their problem and method with the prior work on catastrophic forgetting. It is possible that the problems and their solutions are opposites of each other and would be very useful to study.
- The paper would benefit from more directly exploring their hypothesis of "decaying plasticity" of neural networks by exploring the change in weights, gradients, and loss landscape over time.
- This paper only demonstrated their algorithm on Feed-forward networks. Since the paper is largely empirical in demonstrating this new problem, it would be significantly stronger if it could demonstrate this problem on Deep convolutional models, and attention networks. The model sizes and datasets are also relatively smaller. Does this problem continue with overparameterized models or other regularization techniques.

Questions/Suggesiton:
It may be useful to position the work compared to "Neural Rejuvenation" [1] since the reinitialization method is somewhat similar.

Are the errors shown test errors or training errors? Could this be a problem of overfitting and the proposed method is acting as regularization?

Have you experimented with even lower or higher learning rates? The accuracy for the lowest in the current graphs seems to still converge quite quickly.

Could we compare to a baseline of fully reinitializing the model when the data distribution changes to understand the upper-bound on the expected accuracy?

[1] Qiao, Siyuan, et al. "Neural rejuvenation: Improving deep network training by enhancing computational resource utilization." Proceedings of the IEEE/CVF Conference on Computer Vision and Pattern Recognition. 2019.

**Summary Of The Paper:**

This paper demonstrates and proposes a solution for a new problem in continual learning which is the inverse of catastrophic forgetting. Compared to prior work, they study problems where the data distribution changes much more rapidly. They demonstrate that backpropagation based optimization loses its ability to adapt when tracking these rapidly changing continual learning problems.  They show a degradation in performance over time on permuated MNIST, non-stationary RL problems, and the bit-flipping problem. They propose a solution to this problem by reinitializing some portion of the weights of every layer. They propose a utility function to choose which layers to reinitialize based on a combination of adaptation-utility and contribution-utility. They demonstrate that utilizing this method, they can achieve better performance that does not degrade over time and that it works in more cases than l2 weight decay.

**Summary Of The Review:**

This paper proposes and tackles a very novel problem in continual learning that would have great significance to the field and is important to explore. They are able to demonstrate a somewhat novel effective algorithm that significantly improves performance, and does not degrade over time. Unfortunately, this paper is largely empirical and does not have sufficiently diverse experiments to characterize this problem. Given this, I currently recommend weak rejection for this paper. While it explores a large variety of activation functions and hyperparameters, all the networks are relatively small Feed-forward networks. They compare several methods and demonstrate that regularization is somewhat effective in some cases, but does not fully explore why regularization works or other methods. It currently falls short of fully explaining the root cause of the problem.

=================================================================
Post Rebuttal:
I would again like to thank the authors for including the additional experiments on model size and dataset change speed. I believe this paper is borderline, and am updating my review to marginally above the acceptance threshold. The additional experiments improved understanding and I believe this is a very useful direction to explore. I also agree with reviewer gcKs and Evj3, that this should be compared to existing continual learning methods and papers. The other datasets should also be useful to experiment with since they should still demonstrate at least some loss in Neural plasticity with 20 stationaries according to the other experiments. I also agree with reviewer kzUn on the depth of understanding this paper provides. This paper highlights a very useful novel direction to explore continual learning problems and a reasonable solution, but would drastically be improved if it provided much further depth.

---

> ### Author Response · Authors · 2021-11-12
> **We add some of the suggested studies in the paper**
>
> Thank you for your time and valuable feedback. We are glad to see that you recognize the novelty and importance of establishing decaying plasticity.
>
> A minor correction: We are not focusing on just “rapidly changing continual learning problems.” In the Bit-Flipping problem and Slippery Ant, the distribution changes slowly. It is standard to train PPO for just 3M time-steps on Ant. But, in our case, the environment changes after 10M time-steps. The diversity in the length of training times makes our results more general.
>
> Reply to the weaknesses:
> - We agree that studying the interplay of catastrophic forgetting and decaying plasticity will be an important contribution. However, we think that studying that interplay is beyond the scope of current work. In fact, the same argument could be made that works addressing catastrophic forgetting would be a better contribution if they also studied the interplay of catastrophic forgetting and decaying plasticity or other issues of NNs. Technically, studying more interplays would make those works better, but many works on catastrophic forgetting constitute strong contributions by focusing on the problem singularly.
> - Thank you for pointing this out. We did study the evolution of feature saturation and gradients. We have added a new section in the appendix to show the results.
> - We agree that experiments with bigger networks will make the paper stronger. However, testing if CBP performs well with CNNs, will require a long-term continual learning task with images, which we currently don't have. Most of the existing datasets have a small number of non-stationarities, which are not enough to show the decaying plasticity phenomenon.
>
> Answers to the questions:
> - Neural rejuvenation seems similar to CBP. But, it only reinitializes features that completely die out. So, it is not clear how it will work for tanh. But, let’s assume that it replaces saturated features when used with tanh. In that case, it is similar to Adaptation utility. This utility removes features with the largest input weights, which is similar to removing the most saturated feature. If that is the case, Neural Rejuvenation will perform similar to Adaptation utility which performs significantly worse than overall utility (Figure 17a)
> - No, it is not overfitting because, in permuted MNIST, we only show the images once. So, the online accuracy is the same as the test accuracy. In online learning, there is no notion of training or test phase. There is only online performance and online error
> - For BP, the results are for multiple step-sizes on both Bit-Flipping problem and Permuted MNIST. However, for other algorithms, we used the step-size that performed best with BP. For example, on Permuted MNIST, we used a step-size of 0.003 as it achieved the highest accuracy with BP.
> - In permuted MNIST, the data distribution changes after every 60k examples, and the first data point shows the mean accuracy for the first 60k examples. So, if the model is reinitialized each time the distribution changes, the performance will be the same as that of the first data point (94.2%) in Figure 4.
>
> Reply to the summary of the review:
>
> To address your concern for not having a deeper look into the problem, we have added a new section in the appendix where we look at feature saturation and gradient magnitudes. We used methods like L2 regularization and Normalization because the features get saturated with BP. However, neither of these quantities fully explain the behaviour of all algorithms. So, we didn't include these studies in the original submission. But now, we’ve added those results. They rule out that “feature saturation,” and “low gradients in the input layer,” and “large weight magnitude” as the root cause of decaying plasticity.
>
> With that said, we disagree that "falls short of fully explaining the root cause" can be a basis for rejection. Compelling papers to have a full explanation can cause them to have a wrong explanation, which slows down science. For example, the Batch normalization paper [1] claimed that BatchNorm works because it reduces "Internal Covariate Shift." However, recent work [2] has shown that "Internal Covariate Shift has little to do with the success of BatchNorm." We believe a sustainable and consistent progress in science requires a humble approach to claims, which can be hindered when a full explanation is required of a paper for acceptance.
>
> Our paper aims to establish the existence of “decaying plasticity” and propose algorithms that perform better than BP. Do you not think that we have established both of these points clearly enough? We urge you to consider whether it is the responsibility of one paper or the whole community to explore and explain a new phenomenon fully.
>
> [1] Ioffe, S., & Szegedy, C. Batch normalization: Accelerating deep network training by reducing internal covariate shift. [2] Santurkar, S., Tsipras, D., Ilyas, A., & Mądry, A. How does batch normalization help optimization?

---

> > ### Comment · Reviewer_jf4G · 2021-11-22
> > **Additional questions on the variety of experiments on decaying plasticity**
> >
> > I would like to thank the authors for the inclusion of the experiments on feature and gradient magnitudes. That seems to indicate those do not fully explain this phenomenon.
> >
> > While I agree it is not necessary to explore and explain the root cause of your observed behavior, it would make your paper significantly stronger.
> >
> > I agree that you have robustly demonstrated this behavior in several different tasks. However, the paper would still be significantly stronger if you demonstrated it with a larger variety of models. It would be good to know if this is behavior only observed in feed-forward networks of a certain size and how it is affected by overparameterization Do you happen to have any ablations of model size?
> > How does the speed of the distribution change affect decaying plasticity?

---

> > > ### Author Response · Authors · 2021-11-23
> > > **We added the requested studies**
> > >
> > > Thank you for your reply. We have added the suggested studies in the appendix. Please look at sections F and H.
> > >
> > > The model that we used on Permuted MNIST (3 hidden layers with 2000 hidden units) is overparameterized. Typically feed-forward networks like LeNet [1] (2 hidden layers with 300 and 100 hidden units) perform well on MNSIT. This network has 300k weights, while our network has 10M weights. We added a small study that shows that the performance drop is much faster with a small network.
> > >
> > > Finally, we added a study to look at the effect of the speed of distribution change. It showed that the faster the distribution changes, the faster the performance drops.
> > >
> > > [1] Dettmers, T., & Zettlemoyer, L. (2019). Sparse networks from scratch: Faster training without losing performance. arXiv preprint arXiv:1907.04840.

---

> > > > ### Comment · Reviewer_jf4G · 2021-12-03
> > > > **Thank you for the additional experiments**
> > > >
> > > > First, I would like to thank the authors for so quickly adding these additional experiments. It really helps us explore this behavior of deteriorating neural plasticity. But this behavior could still be much better explored in this paper. It could be a slight problem in the presentation, but it's not entirely clear if the decaying Neural plasticity is the network losing the ability to adapt completely to new tasks, or the ability to adapt quickly to new tasks. The network with the significantly slower change seems to indicate that it is mostly the ability to adapt quickly since with repetitive training on the exact same data, it is still able to fit the data. Though it should be noted that the experiment is fundamentally a different task than the other examples since it saw the same training data multiple times compared to the other training runs.
> > > >
> > > > I believe this paper is borderline, and I will update my review to marginally above the acceptance threshold since the additional experiments improved understanding and I believe this is a very useful to direction to explore. It would be significantly stronger if the behavior and mechanism was significantly better understood and characterized and as reviewer gcKs noted, it was compared to existing works to explore if they improved or worsened performance.

---

### Official Review · Reviewer_gcKs · 2021-11-07

**Correctness:** 4
**Technical Novelty And Significance:** 3
**Empirical Novelty And Significance:** 3
**Recommendation:** 5
**Confidence:** 5

**Main Review:**

* The shortcomings of SGD have motivated many works on its adaptation for continual learning and several adaptations such as Orthogonal Gradient Descent [1], Stable-SGD [2] and alternative local learning approaches [3,4] have been proposed. A comparison with these methods would help with better assessing the proposed approach.

* Other benchmarks such as split-CIFAR100, split-IMAGENET, CoRe50 are frequently used to benchmark the continual supervised learning approaches. It would be fair to expand the comparison to a subset of these datasets.

* Classification accuracy might not be the best metrics to evaluate the continual leaning potential. Other metrics such as the backward and forward transfer are better suited [5].

[1] Farajtabar, Mehrdad, et al. "Orthogonal gradient descent for continual learning." International Conference on Artificial Intelligence and Statistics. PMLR, 2020.
[2] Mirzadeh, Seyed Iman, Mehrdad Farajtabar, and Hassan Ghasemzadeh. "Dropout as an implicit gating mechanism for continual learning." Proceedings of the IEEE/CVF Conference on Computer Vision and Pattern Recognition Workshops. 2020.
[3] Lindsey, Jack, and Ashok Litwin-Kumar. "Learning to learn with feedback and local plasticity." arXiv preprint arXiv:2006.09549 (2020).
[4] Madireddy, Sandeep et al. “Neuromodulated Neural Architectures with Local Error Signals for Memory-Constrained Online Continual Learning”, arXiv preprint arXiv:2007.08159 (2021)
[5] Mai, Zheda,  et al. "Online Continual Learning in Image Classification: An Empirical Survey." arXiv preprint arXiv:2101.10423 (2021).


**Summary Of The Paper:**

This work highlights the shortcomings of the backpropagation algorithm for continual learning applications and proposed continual backprop that used a generate-and-test method to continually inject random features alongside SGD which enables better learning on non-stationary data streams. This approach is evaluated with the semi-stationary scenario of Bit-Flipping and continual supervised learning with permuted-MNIST as well as on non-stationary RL problems.

**Summary Of The Review:**

The approach looks promising, but is limited in terms of the experiments and the comparison with existing approaches that makes it difficult to assess the true potential of this approach

---

> ### Author Response · Authors · 2021-11-12
> **The suggested methods are not competitors with CBP, and the suggested datasets are not suitable for our purpose**
>
>
> Thank you for your time and review.
>
>
> - We do not think that comparing CBP with the methods that you proposed is relevant. Because the methods you listed are not competitors with CBP, if anything, we should use them in conjunction with CBP. They are not competitors simply because those methods are solving an orthogonal issue to CBP. They are a solution for "catastrophic forgetting" (stability-plasticity dilemma), while CBP is a solution for "decaying plasticity." To give an analogy from RL, it will be like comparing UCB [1], an exploration algorithm, with DYNA [2], a planning algorithm. Yes, both are part of RL, but they're solving orthogonal issues and are never compared in the literature. Similarly, the methods you suggested and CBP are part of continual learning, but they are solving different issues.
> - The datasets that you suggested are not suitable for our purposes. In our paper, we established the decaying plasticity of Neural Networks. The decaying plasticity phenomenon is clearly observed only when there are many non-stationarities (sub-tasks), and learning happens for an extended period. But, the datasets that you suggested have very few non-stationarities. For example, split-CIFAR 100 has just 20 non-stationarities.
> - In our opinion, the online metric [3] is the ideal way to measure the performance of learning systems in non-stationary problems. Measuring performance on previously seen data can be helpful sometimes because the old data might recur, not because we want to do good on old data. Doing well on old data is only valuable if the old data can recur in the life of the learning system, and if it recurs, then the online loss will measure performance on it when it does recur.
>
> Now that we have explained that we are not missing any comparison with existing approaches or benchmarks on standard datasets, we hope we have addressed all of your concerns.
>
>
> [1] Auer, P. (2002). Using confidence bounds for exploitation-exploration trade-offs. Journal of Machine Learning Research, 3(Nov), 397-422. [2] Sutton, R. S. (1991). Dyna, an integrated architecture for learning, planning, and reacting. ACM Sigart Bulletin, 2(4), 160-163. [3] Orabona, F. (2019). A modern introduction to online learning. arXiv preprint arXiv:1912.13213.

---

### Decision · Program_Chairs · 2022-01-20

**Decision:**

Reject

**Comment:**

The paper studies an important newly identified problem in continual learning of rapid adaptation, and proposes the use of a generate-and-test method to continually inject random features alongside SGD, enabling better learning on non-stationary data streams.
Unfortunately the paper remained borderline in the discussions. While reviewers liked the overall research direction and contributions, they also agreed the paper in current form still would benefit from deeper insights into the proposed method, stronger empirical evidence.
Experiments cover broad applications including RL, but don't seem to give very clear advantages over other weight regularization schemes, and other metrics of quality could be added. We appreciate the authors have added additional experiments testing it both for the two important regimes of under- and over-parameterized networks, though those can be expanded.
We are sorry that this good paper remained narrowly below the bar in this case, and hope the detailed feedback helps to strengthen the paper for a future occasion.